# Modeling diffusive search by non-adaptive sperm: Empirical and computational insights

**Benjamin M. Brisard[1], Kylie D. Cashwell, Stephanie M. Stewart[1], Logan M. Harrison[1], Aidan C. Charles[1], Chelsea V. Dennis[1], Ivie R. Henslee[1], Ethan L. Carrow[1], Heather A. Belcher[2], Debajit Bhowmick[3], Paul W. Vos[4], Maciej Majka[5], Martin Bier[6], David M. Hart[7], Cameron A. Schmidt[1]**

**1** Department of Biology, East Carolina University, Greenville, North Carolina, United States of America, **2** Department of Anatomy and Cell Biology, Brody School of Medicine, East Carolina University, North Carolina, United States of America, **3** Flow Cytometry Core Facility, Brody School of Medicine, East Carolina University, North Carolina, United States of America, **4** Department of Public Health, East Carolina University, Greenville, North Carolina, United States of America, **5** Institute of Theoretical Physics and Mark Kac Center for Complex Systems Research, Jagiellonian University, Krakow, Poland **6** Department of Physics, East Carolina University, Greenville, North Carolina, United States of America, **7** Department of Computer Science, East Carolina University, Greenville, North Carolina, United States of America

* schmidtc18@ecu.edu

## Abstract

During fertilization, mammalian sperm undergo a winnowing selection process that reduces the candidate pool of potential fertilizers from ~$10^6$-$10^{11}$ cells to $10^1$-$10^2$ cells (depending on the species). Classical sperm competition theory addresses the positive or 'stabilizing' selection acting on sperm phenotypes within populations of organisms but does not strictly address the developmental consequences of sperm traits among individual organisms that are under purifying selection during fertilization. It is the latter that is of utmost concern for improving assisted reproductive technologies (ART) because low-fitness sperm may be inadvertently used for fertilization during interventions that rely heavily on artificial sperm selection, such as intracytoplasmic sperm injection (ICSI). Importantly, some form of sperm selection is used in nearly all forms of ART (e.g., differential centrifugation, swim-up, or hyaluronan binding assays, etc.). To date, there is no unifying quantitative framework (i.e., theory of sperm selection) that synthesizes causal mechanisms of selection with observed natural variation in individual sperm traits. In this report, we reframe the physiological function of sperm as a collective diffusive search process and develop multi-scale computational models to explore the causal dynamics that constrain sperm fitness during fertilization. Several experimentally useful concepts are developed, including a probabilistic measure of sperm fitness as well as an information theoretic measure of the magnitude of sperm selection, each of which are assessed under systematic increases in microenvironmental selective pressure acting on sperm motility patterns.

**Data availability statement:** All data and code used for running experiments, model fitting, and plotting is available on a GitHub repository at https://github.com/cas-mitolab/Fertilization_ABM

**Funding:** This work was supported by the Eunice Kennedy Shriver National Institute of Child Health and Human Development (R01HD110170 to CAS), as well as laboratory startup funding from the Thomas Harriot College of Arts and Sciences at East Carolina University and the East Carolina University Research and Economic Development Office (CAS). M.M. gratefully acknowledges the support for this research by Fulbright Scholar-In-Residence Program, sponsored by the U.S. Department of State. The funders had no role in study design, data collection and analysis, decision to publish, or preparation of the manuscript.

**Competing interests:** The authors have declared that no competing interests exist

## Author summary

During mammalian reproduction, sperm outnumber eggs by many orders of magnitude. This study models the statistical properties of sperm movement as a diffusive search process, combining experiments and simulations to explore how heterogeneity in motility patterns and microenvironmental complexity shape successful fertilization. We introduce simple metrics to quantify sperm fitness and the magnitude of selection pressure imposed by the microenvironment, revealing that sperm phenotype distributions interact with environmental constraints to determine the range of sperm traits that ultimately support successful egg contact. These insights improve the understanding of sperm subpopulation dynamics and offer practical tools for optimizing assisted reproductive technologies in clinical and agricultural settings.

## Introduction

Assisted reproductive technologies (ARTs) are widely used in medicine and agriculture and include a variety of strategies such as *in vitro* fertilization, intra-uterine insemination, and embryo transplantation. Efficiency of ART is of utmost importance because of the implications for parental and offspring well-being, and significant time and cost investments. Though there are a multitude of factors that influence ART efficiency, one particularly salient challenge has been the pre-selection of sperm that have the potential to maximize the paternal contribution to the number and quality of viable embryos [1–3]. Identifying and isolating sperm with high fertility and developmental potential presents a significant challenge due to their structural and phenotypic heterogeneity, dynamic post-ejaculatory maturation processes, and the large quantity of cells in an ejaculate (i.e., order of $10^6$-$10^{11}$ depending upon species) [4–10].

Phenotypic variation in sperm has generally been explained by a game-theoretic competition model in which males adopt evolutionarily stable strategies that maximize fitness payoffs under sexual selection [11]. For example, mammalian sperm exhibit relatively high swimming velocity and/or greater sperm number per ejaculate in socio-ecological scenarios where there is strong between-male competition for mates [12]. Largely inspired by those observations, swimming velocity and sperm count have been regarded as heuristic guides for clinical sperm selection under the straightforward assumption that the 'highest quality' sperm can be identified from an idealized set of competitive traits [13].

However, heuristic approaches may be misleading because the predictions of sperm competition theory apply only to *between-male* variation, while *within-male* variation in sperm phenotype is the primary concern of assisted reproduction [11,14]. Importantly, male gamete function not only co-evolves with the competitive traits of other males, but also with the corresponding micro/macro-scale anatomy of the female reproductive tract. This effect, known as 'cryptic female choice', facilitates sperm selection in the reproductive microenvironment through various physical and chemical barriers (e.g., epithelial folds, cervical mucous, etc.) [15]. For example, mammalian sperm have evolved time-dependent changes in motility pattern (e.g., progressive to hyperactivated transition) that assist navigation of the labyrinth-like epithelial surfaces of the oviducts [16]. *Within-male* sperm selection may be an important component of mammalian reproduction and is a powerful candidate for the improvement of ART outcomes. However, our understanding of sperm selection at the cell population scale remains limited, and there is currently no underlying theory that enables precise description of the key aspects of sperm selection - including a quantitative definition of sperm 'fitness', or a measure of the magnitude of selective pressure acting on sperm traits under a given set of conditions.

In this report, we investigate sperm selection as a consequence of the interaction between phenotypic variation among sperm populations and the constraints imposed on sperm fitness by the reproductive microenvironment. We use empirical data to inform the development of agent-based computational models (ABMs) and simulate 'bottom up' sperm population dynamics. We then extend concepts from probability and information theory to define a quantitative measure of sperm fitness (i.e., the posterior probability distribution of 'successful' traits obtained using Bayes theorem), as well as a measure of the magnitude of selection imposed on a sperm population during fertilization (i.e., the relative information gain). The results from this work lay a foundation for high-precision male fertility diagnostics to improve sperm classification and/or selection in conjunction with existing semen analysis and laboratory pre-selection procedures.

## Results

### Model aim and context

The models are meant to simulate physiologically relevant aspects of sperm motility that contribute to sperm selection under microenvironmental constraints. The core models were designed using simple self-propelled particle physics, similar to the methods used in Computer Aided Sperm Motility Analysis (CASA), in which motility measures are obtained by digitally locating, annotating, tracking, and summarizing the trajectories of sperm nuclei in microscopy videos [17]. The models were fit to empirical data to improve accuracy and physiological relevance.

### System boundaries

The physical environment simulated by the models approximates a 10X field-of-view under a light microscope with 680 X 680 μm side lengths and approximately $4.62 \times 10^5$ μm² area (Fig 1A). Sperm motility imaging is typically performed using ~20 μm depth chambered slides that restrict the axial mobility of the cells for the study of 'planar' flagellar beating [18]. Although the models are 2D, they can be considered to have a 3D quality because the sperm are allowed to freely cross paths, as would occur in depth-chambered slides. The environment is comprised of a regular grid-space arranged in four quadrants. The grid squares were assigned a length scale value chosen to facilitate accurate approximation of empirically derived microscopy data (40 μm for the dimensions described above) [19]. The length scale factor can be adjusted to facilitate modeling any spatially defined environment.

### Core movement functions

The agents in the simulations represent the spermatozoon nucleus, in the same manner that sperm nuclei are typically imaged, filtered, and tracked using phase contrast microscopy for 2D path reconstruction in computer aided sperm motility analysis (CASA) [17].

The agents in the simulations execute the self-propelling random walk (Fig 1B). That is, each agent is characterized by its velocity vector $\vec{v}_i(t) = (\dot{x}_i(t), \dot{y}_i(t))$ and the angle $\theta_i(t)$ at which this velocity vector is directed, subjected to random fluctuations. The components of $\vec{v}_i(t)$ and $\theta_i(t)$ obey the following equations of motion:

$$\dot{x}_i(t) = \cos(\theta(t))[v + \sigma_r \eta_{r,i}(t)]$$
$$\dot{y}_i(t) = \sin(\theta(t))[v + \sigma_r \eta_{r,i}(t)]$$
$$\dot{\theta}_i(t) = sgn(\sin(2\pi t / \tau))[\omega + \sigma_\theta \eta_{\theta,i}(t)]$$

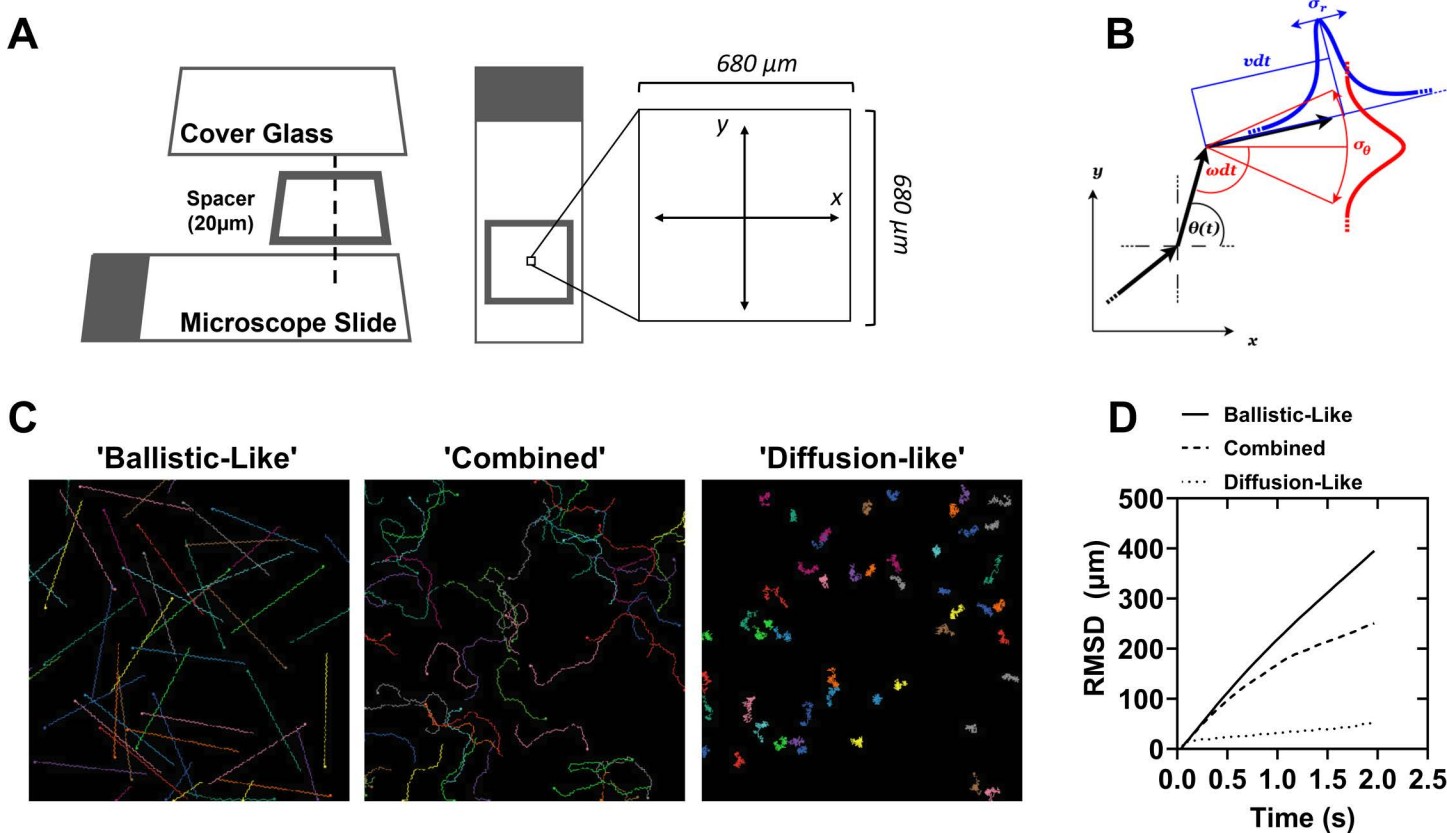

**Fig 1. Simulated random walkers explore space in a manner that depends on their movement properties.** (A) Diagram of the model environment, which is designed to emulate an isolated portion of the field of view of a light microscope with a 10X objective and a 20 μm deep chambered glass microscope slide. (B) Diagram of the core movement functions employed by the agent-based model. Θ(t) is the angular rate of change, v(t) is the radial rate of change, σ is the respective amplitudes of zero average Gaussian noise added to the parameters. (C) Example images of simulations highlighting extremes of model behavior based on the choice of parameters. Ballistic-like motion results from no Gaussian noise being added to the radial and angular velocities; Combined motion results from combinations of the radial and angular velocities as well as the amplitude of noise added to each term; Diffusion-like motion results from relatively large values of noise amplitude. (D) Root mean square displacement of simulations with 50 agents as a measure of the relative distance traveled by the particles on average from their point of origin after 50 steps. Colors are randomly assigned to the agents and serve only to facilitate distinguishing the trajectories.

Here, $T$ numerates the agents, while $\eta_{r,i}(t)$. and $\eta_{\theta,i}(t)$ are the zero-average uncorrelated Gaussian noises and $sgn(\ldots)$ is the signum function. When random fluctuations are not present ($\sigma_r = 0$ and $\sigma_\theta = 0$), these equations describe a trajectory consisting of alternating arcs, oscillating around a preset direction. In this case, $\sigma$ describes the magnitude of self-propulsion, $N$ is the period of oscillations in time and the particle turns its velocity vector by the angle $\omega\tau/2$ as it travels along a single arc of its trajectory. When fluctuations are present ($\sigma_r \neq 0$ and $\sigma_\theta \neq 0$), $\sigma$ can be interpreted as the average velocity $|\vec{v}(t)|$ and $sgn(\sin(2\pi t/\tau))\omega$ is the average angular velocity $\dot{\theta}_i(t)$. The presence of fluctuations introduces random direction changes into the motion of the agents. Depending on the magnitude of fluctuations (given by $\sigma_r$ and $\sigma_\theta$) and the specific choice of $\sigma$ and $m$, a range of widely different movement patterns can be described by the above model.

Two extreme examples of simulated movement patterns, spanning the conceptual spectrum of possible behaviors (Fig 1C), are ballistic-like motion and free diffusion. Choosing $\sigma_r \ll v$ and $\sigma_\theta \ll \omega$ while keeping non-zero $r$ and $\omega$ such that $0 < \omega\tau/2 < \pi$, results in the trajectories resembling the deterministic alternating arcs pattern – or ballistic-like motion. In

this case, the agents effectively travel long distances, but do not eensively explore the surroundings of their current position. In case of $\omega\tau/2 > \pi$ the paths become circular, which can be seen as a quasi-deterministic strategy for more local search. However, the ballistic-like trajectories are generally 'stiff', that is, they are characterized by long-persisting correlations in the changes of position. Conversely, assuming $v = 0$ and $\omega = 0$ results in motion akin to passive diffusion. In this case, the agents thoroughly explore their immediate surroundings, but on average do not change position. The correlations in position change are also very short-ranged. The difference between both archetypes of movement pattern is conveniently characterized by the root mean squared displacement:

$$RMSD_i(t) = \sqrt{\left(x_i(t) - x_i(0)\right)^2 + \left(y_i(t) - y_i(0)\right)^2}$$

For ballistic-like motion (with $0 < \omega\tau/2 < \pi$) $RMSD \propto t$, while for free diffusion $RMSD \propto \sqrt{t}$, where in the former case RMSD is the measure of travelled distance, while in the latter case it is a radius of explored area (Fig 1D). In general, the model (1) can describe a continuity of possible motility patterns. By varying the parameters of the model (especially, by allowing $\sigma_\theta$ to be comparable with $\omega$, while keeping $v \gg \sigma_r$) it is possible to combine the aspects of ballistic-like motion and free diffusion in almost arbitrary proportions. Significant variety of movement patterns is observed in mammalian sperm, making the proposed movement functions particularly suitable as a model of sperm motility [19,20].

## Timescale of the models

Time in the models advances in discrete steps during which agent states are independently updated in random sequence (i.e., asynchronously). Model-time was scaled to real-time by setting the timestep of an advancement to previously reported beat cross-frequencies for isolated mouse sperm [19]. For example, reported mean beat-cross frequency is 25.4 Hz, and each sperm crosses its average straight-line path once per model-time advancement, then one advancement is approximately 1/25.4 seconds or approximately 40 milliseconds. The model length and time scales can be readily adjusted to simulate behavior of sperm from species other than mouse, but mouse parameters are used throughout this report for consistency with available data.

## Parameter estimation for the sperm motility patterns

Isolated mouse epididymal sperm have been studied extensively to define the molecular mechanisms and phenotypic characteristics of fertilization competent sperm. Though there is a large set of possible movement characteristics that a given sperm may occupy at a given time, some common features of motility can be classed into specific patterns. For example, 'progressive' motility consists of a symmetric movement with low lateral head amplitude about the averaged central path and rapid straight-line movement [17]. Similarly, 'intermediate' motility follows a similar movement pattern, but with greater magnitude of lateral head displacement. Here, we define five categorical motility patterns, based on previous work [19].

Typical CASA motility parameters consist of: VAP (average path velocity), VSL (straight line velocity), VCL (curvilinear velocity), ALH (amplitude of lateral head), BCF (beat-cross frequency), STR (straightness), and LIN (linearity) [17]. VSL, VCL, and VAP were used to compare the parameter fitting of the sperm movement functions in our models to empirical sperm motility data. Means and standard deviations of the movement parameters for each motility type were obtained from a previous report, and Gaussian distributions were simulated using Graphpad Prism [19]. Temporally-coded phase contrast images of representative cauda epididymal mouse sperm movement patterns at 10X magnification are qualitatively similar to those observed for

sperm (shown at approximately 10X magnification at the objective; Fig 2A). Movement function parameters (VSL, VCL, and VAP) for the agents of each prescribed motility class were adjusted to approximate the medians of the distributions of sperm movement parameters (Fig 2B–D). The movement function parameter values are detailed in (Table 1). The distributions did not match exactly for every parameter or motility pattern, indicating that true sperm motility is more complicated than the simple agent rules in this report. Nevertheless, the simulated motility patterns are qualitatively similar to mouse sperm and could be readily updated in future model iterations to accommodate alternative agent rulesets, or movement parameter distributions from

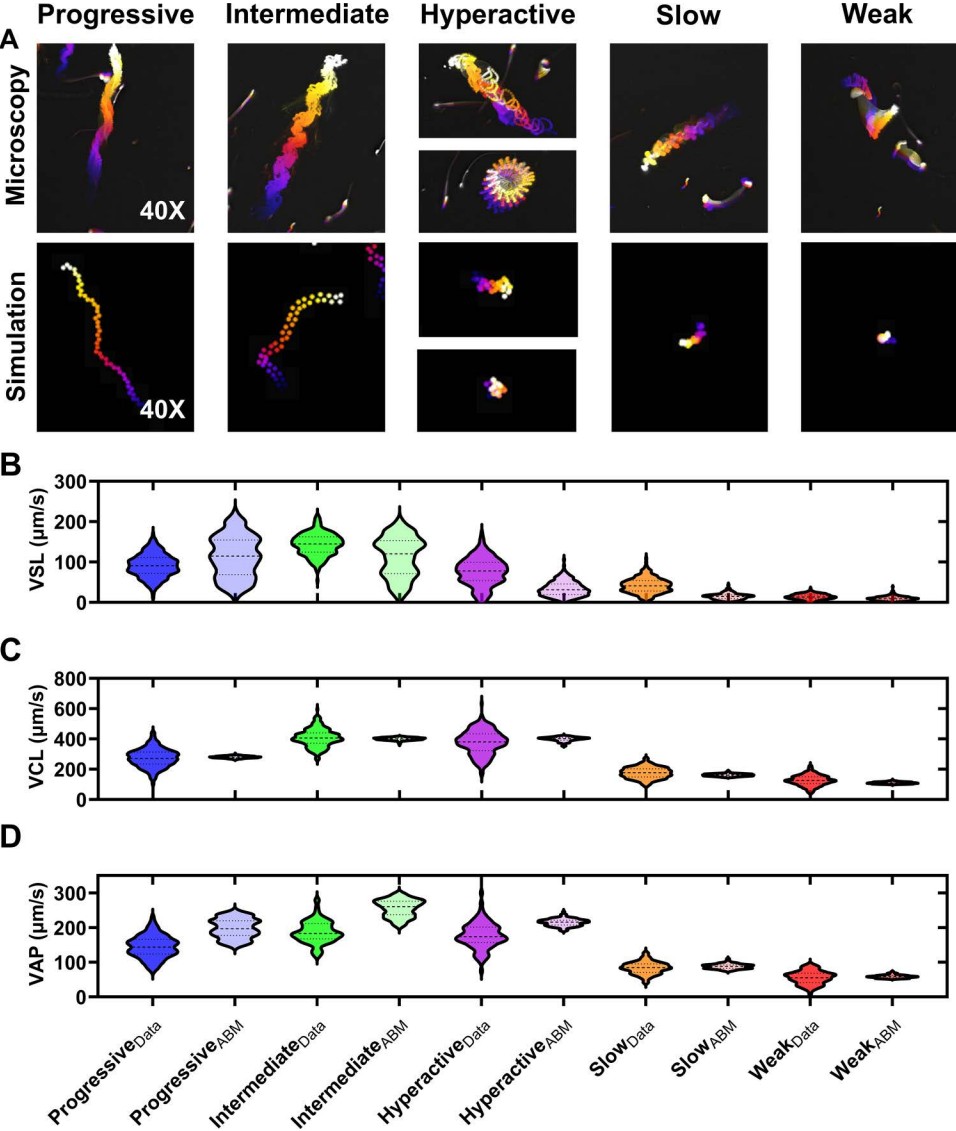

**Fig 2. Parameter Estimation for the Sperm Motility Patterns.** (A) Temporally coded phase contrast images of mouse sperm motility patterns (top row). Matching temporally coded images of simulated sperm-agents for each sperm motility type (bottom row). Color scale is blue (early) to white (late) frames in the video. Simulated trajectories (B) Consensus data from published studies were used to generate normal distributions of curvilinear velocity (VCL) values (indicated by the subscript 'Data'). Parameters (i.e., $v(t)$, $\theta(t)$, $\sigma r$, and $\sigma_\theta$) of each motility type in the agent-based models (i.e., subscript- ABM) were adjusted to approximate the mean VCL values with those identified in the data distributions. **N** = 250 data points for all groups.

**Table 1. Movement function parameters for sperm simulations.**

| Motility pattern | Nominal Radial Velocity $v\left[\mu m/s\right]$ | Std. Deviation of Radial Gaussian Noise $\sigma_r\left[\mu m/s^{1/2}\right]$ | Nominal Angular Velocity $\omega\left[°/\tau\right]$ | Std. Deviation of Angular Gaussian Noise $\sigma_\theta\left[°/\tau^{1/2}\right]$ |
|---|---|---|---|---|
| Progressive | 279.4 | 0.58 | 45 | 17.3 |
| Intermediate | 419.1 | 0.29 | 120 | 11.5 |
| Hyperactive | 419.1 | 0.29 | 180 | 51.0 |
| Slow | 152.4 | 1.15 | 135 | 26.0 |
| Weak | 101.6 | 0.29 | 180 | 52.0 |

other species if labeled CASA data are available. For instances where labeled data is not available, meaning that categorical motility types are not classified, the models could still be fit directly to measured CASA parameter distributions (e.g., VCL, VSL, ALH, etc).

Parameter to control the oscillation period of the signum function: $\tau = 1/25.4\ [s]$.

## Sperm-agent search is a function of ensemble motility pattern

We performed simulations and sensitivity analysis to investigate the relationships between motility pattern and the amount of discrete space searched in a bounded environment. Each simulation was performed with agents of only one motility type (progressive, intermediate, hyperactive, slow, weak, or mixed in equal proportions). Sperm began in a randomized position in the environment with the same total number in each simulation (N = 250). The environment consisted of an underlying grid of 40 x 40 μm unit squares. Each grid square was considered distinctly searched if at least one sperm passed through it. Simulations ended after two seconds, a typical time window for computer aided sperm motility analysis (CASA). The time color-coded movement paths of a representative simulation with 'mixed' sperm motility types are shown (Fig 3A). Populations of sperm with mixed motility types exhibited average movement patterns and an ability to search space that was reflective of the proportions of different motility types in the population (Fig 3B,C). These results demonstrate that the *average* search capability of a sperm population reflects the underlying distribution of sperm motility phenotypes. Notably, there are important mathematical properties exhibited by correlated random walks that may be useful for improving sperm motility analysis and modeling the process of fertilization more broadly (see the discussion section for more details).

## Adding spatial complexity to the microenvironment

The microanatomy of mammalian female reproductive tracts imposes spatial limitations on sperm movement that act as physical barriers to eventual contact with the egg(s). In the uterus, the luminal volume is large relative to the size of a single sperm, and convective flow predominates in the dispersion of cells [21]. However, the luminal volume in the cervix and oviducts are much smaller relative to the size of a sperm and the relative volumes are constrained by the presence of laminar epithelial folds which form a tight labyrinth-like environment [22].

To model the relationship between microenvironmental complexity and sperm selection, three simulation environments were developed. Mazes (more specifically labyrinths, or 'acyclic' mazes) were chosen as a simple model of microenvironmental spatial complexity because they can be compared quantitatively using foundational concepts from graph theory. These simple structures are not necessarily intended to serve as accurate models of oviducts, which are much more complex and involve adaptive physiological variables including hormones,

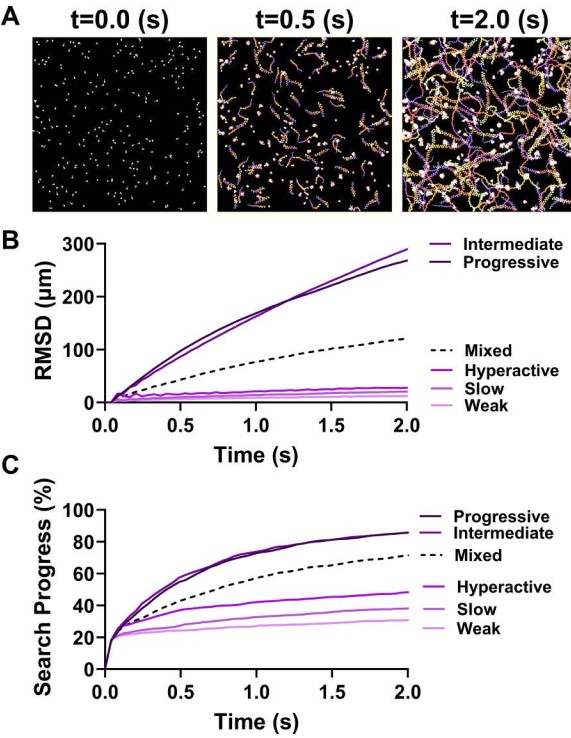

**Fig 3. Sperm-Agent Search is a Function of Ensemble Motility Pattern.** (A) Representative model simulation of 250 sperm with equal proportions of each motility type searching a closed space. Color scale is blue (early) to white (late) frames in the video. (B) Root mean squared displacement (μm) for simulations involving the indicated composition of motility types. Mixed populations consisted of 50 sperm of each motility type. (C) Search progress (%) for the simulations described in subpanel (B).

metabolites, hydrodynamic forces, and thermal gradients. Rather, the simple mazes enable quantitative exploration of the fundamental constraints imposed on sperm fitness by a selective pressure (in this case- spatial complexity), and can be more readily mimicked *in vitro*, making them more tractable for experimental validation using channel-based microfluidics.

The mazes in this report were defined by internal (white) barriers that the sperm could not cross. A separate agent-based model was designed to facilitate drawing and saving the maze environments (available at https://github.com/cas-mitolab/Fertilization_ABM). When a sperm encountered a barrier, the sperm would reorient within a range of possible new directions determined by their motility state and corresponding movement function. A single 'egg' was also included in the environment as a designated grid square, and egg-contact occurred when a sperm moved over the square. The mazes consisted of 'dead-ends' and 'intersections' as vertices of an undirected graph $G(v,e)$, where $v$ is a set of vertices $\{v_i\}$ and $(e)$ is a set of edges where $e_{ij}$ *is the unordered vertex pair* $\{v_i, v_j\}$. We define a path as the set of edges that connects two specified vertices. To quantify the complexity of the mazes, the total weighted complexity ($TC_w$) was calculated as:

$$TC_w = \sum_{i=1}^{k} d_i$$

where $d_i$ is the i$^{th}$ vertex degree (i.e., the total number of paths that lead to and from the vertex) and the vertex indices $\{1, 2, …, k\}$ is a proper subset of $v$. An open space in which sperm start at one position and an egg is located at another position within the space, has a total weighted

complexity of 1 (Fig 4A). Mazes with more vertices, or with more edges connecting vertices are more complex and reflect a larger $TC_w$ (Fig 4B,C). As an additional measure of spatial complexity, we considered the probability of a sperm traversing a direct path from the start (S) position to the (E) egg position, which can be calculated as:

$$P(S \rightarrow E) = \prod_{i=1}^{k} \frac{1}{d_i}$$

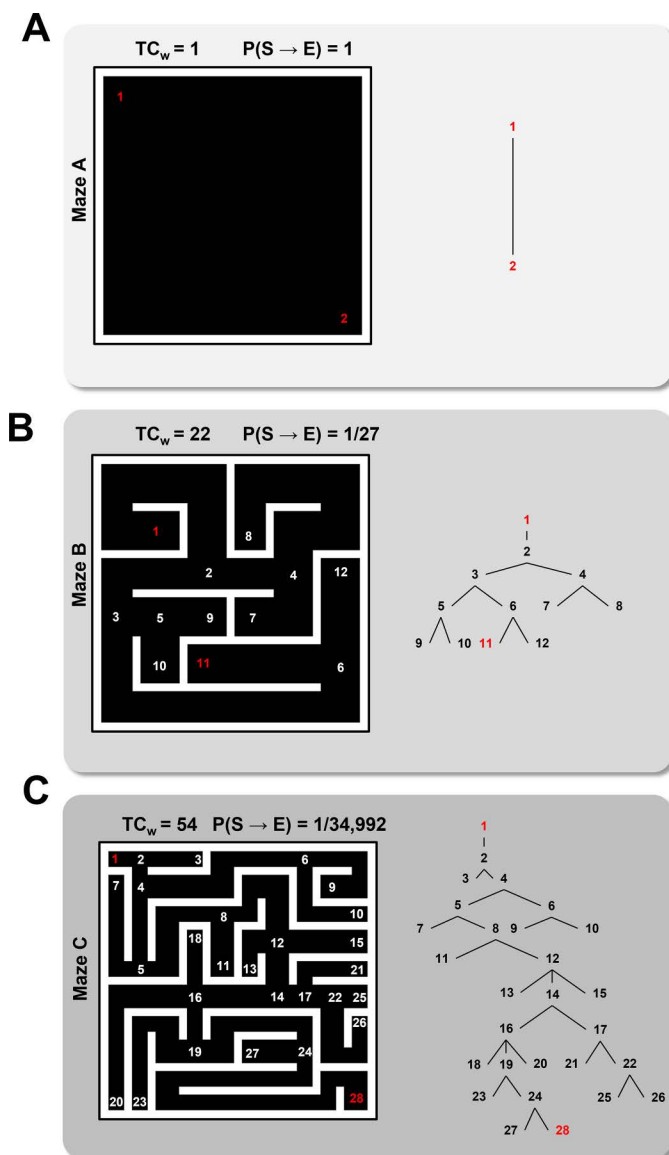

**Fig 4. Modeling Microenvironmental Complexity.** (A) The simplest simulation microenvironment consisting of an open space with an egg located in the bottom right corner. Sperm begin at position 1 (red) and end at the egg position 2 (red). TCw = total weighted complexity, a measure of the graph complexity of the maze. P(S→E) = the probability of a sperm taking the shortest direct path to the egg. (B) A more complex maze with increased TCw relative to maze A. (C) The most complex maze used in the simulations. Mazes were constructed using a separate agent-based model in Netlogo. Vertex numbers are indicated on the maze diagrams. Graph networks with numbered vertices connected by edges are shown on the right.

where k is the total number of vertices along the shortest path from the sperm to the egg. For example, the probability of a direct path taken (Fig 4C) is extremely low (i.e., 1/34,992 possible paths).

## Sperm number and search properties

To explore the relationship between sperm-agent number (or density within the simulation space) and search for an egg in complex spatial microenvironments, simulations were performed for increasing numbers of sperm (1-$10^4$ progressively motile sperm; 100 simulations each). The simulations ended when the egg was contacted for the first time by one of the sperm (Fig 5A; top- $TC_w = 1$; middle- $TC_w = 22$; bottom- $TC_w = 54$). Notably, the time to first contact was not symmetrically distributed for each sperm-agent density, which became symmetric when placed on a logarithmic scale. Simulations involving $10^3$ and $10^4$ sperm became much more likely to be normally distributed rather than lognormally distributed in Mazes B and C, but not A. However, tests for lognormality were inconclusive and it is not clear what the underlying distributions were in either case, though it was clear that the distributions were skewed in simulations with fewer sperm (Fig 5A).

Next, we investigated the relative impact of environmental complexity on the efficiency of the search process by examining what quantity of the searchable area was accessed during each simulation (Fig 5B; top- $TC_w = 1$; middle- $TC_w = 22$; bottom- $TC_w = 54$). We hypothesized that sperm ensembles would perform something like a depth-first search in which

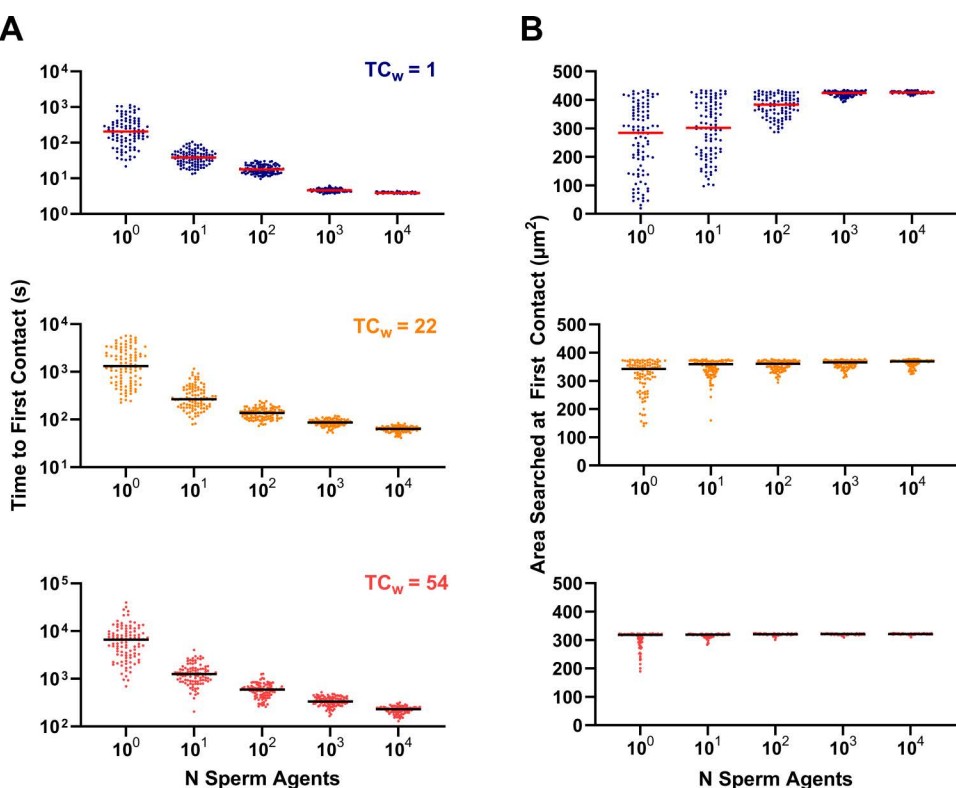

**Fig 5. Sperm Number and Search Properties.** (A) Time to first contact with an egg in microenvironments with TCw = 1 (maze A), TCw = 22(maze B), and TCw = 54 (maze C). (B) Area (μm²) searched at first contact with the egg for microenvironments with increasing weighted complexity (top to bottom as in subpanel A). Lines indicate the median. **N** = 100 simulations for each condition.

all preceding branches of the maze were likely to be searched prior to finding the egg [23]. Indeed, the sperm searched more space prior to finding the egg as the number of agents in the simulations increased (Fig 5B). Interestingly, at low sperm number ($< \sim 10^3$), the role of chance was large compared to simulations with larger numbers of sperm, and in many cases the sperm were able to contact the egg without searching a large proportion of the space. This effect was particularly relevant in the open environment of maze A, but was diminished with increasing environmental complexity in mazes B and C.

Taken together, these results predict that diffusive search for an egg by non-adaptive sperm will exhibit a non-linear relationship with sperm density and that the role of chance in finding shortest path to the egg is modified by the spatial complexity of the microenvironment. As the microenvironment becomes more complex, more sperm are required to minimize the time to egg contact, but the benefit gained by increasing sperm number above a critical threshold also diminishes nonlinearly due to convergence on the most direct path to the egg. These insights may provide a basis for optimal prediction of sperm number for ART procedures such as IVF, though these models do not explicitly account for the risk of polyspermy which would likely form an upper bound on sperm density due to decreasing zygote fitness with increased risk of polyspermy. Additionally, there are several useful mathematical properties that describe the asymptotic behaviors of persistent random walks in mazes that may inform sperm motility analysis and the physiology of fertilization more broadly (see the 'random walks' subsection the discussion for more details).

## A time homogeneous markov model of sperm phenotype heterogeneity

Individual sperm undergo dynamic changes that are conditioned on nutrients and signaling factors in the microenvironment. These factors ultimately influence sperm behavior, lifespan, and ability to recognize and bind to an egg [5,24–27]. In mammalian sperm, intracellular calcium is a key second messenger that mediates capacitive changes, and heterogeneity in calcium transients are a key source of individual sperm variation within cell populations [9,28]. To aid in choosing parameter distributions for the agent-based models, we explored the effect of natural variation in isolated mouse cauda epididymal sperm in response to well-defined capacitating signaling inputs. To account for the controlling effect of exogenous free calcium, we performed pseudo-titrations of total calcium using an ethylene glycol-bis(β-aminoethyl ether)-N,N,N′,N′-tetraacetic acid (EGTA) chemical 'clamp' system to buffer the free calcium at defined concentrations (Fig 6A). We then combined this method with pseudo-titrations of sodium bicarbonate ($HCO_3^-$), a key signaling factor that stimulates capacitation via activation of soluble adenylate kinase. Intracellular calcium was monitored using an acetoxy-methyl ester Indo-1 (ratiometric) dye. $HCO_3^-$ stimulated intracellular calcium increase concomitant with exogenous free calcium concentration over a period of two hours (Fig 6B). These measurements highlight the average responses that sperm populations make to signaling inputs during *in vitro* capacitation.

Next, we sought to determine how intracellular calcium was distributed among individual sperm at the 60-minute timepoint using spectral flow cytometry, with a similar multi-dimensional culture array scheme (Fig 6C). Examination of the qualitative distributions of intracellular calcium ($[Ca^{2+}]_i$ with increasing concentrations of $HCO_3^-$ revealed that $[Ca^{2+}]_i$ exhibited a positively skewed distribution. We interpreted the skewed shape of this distribution as an indication that high $[Ca^{2+}]_i$ cells are a relatively 'rare' phenotype relative to the mean, a pattern which was invariant to the magnitude of the $HCO_3^-$ signal (Fig 6C).

To incorporate this information into updated models with physiological changes in motility phenotype over time, the sperm-agents were updated to include a 'calcium oscillator'

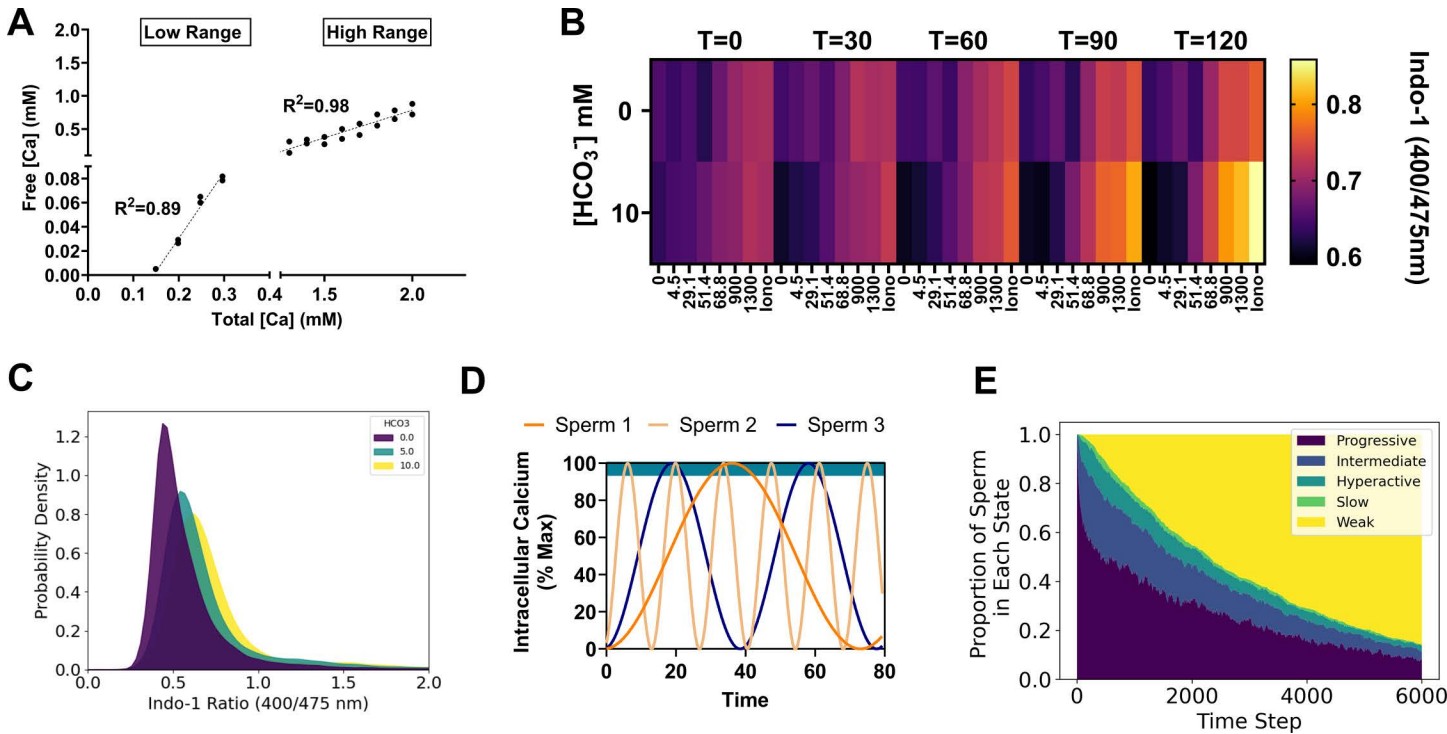

**Fig 6. A Time-Homogeneous Markov Model of Sperm Phenotype Heterogeneity:** (A) Linear regression curves for different calcium ion selective electrode filling solutions used to calculate the free Ca2+ concentrations in HEPES buffered assay media in the presence of 1mM EGTA. (B) Representative heat map showing Indo-1 fluorescence ratios for sperm under the indicated Ca2+ and HCO3- pseudo-titration conditions. Iono = ionomycin. Free calcium concentrations (bottom) are in micromolar units. **T** = time since the beginning of the assay in minutes. (C) Probability density estimate from spectral flow cytometry for approximately 105 live cells per indicated condition. Dead cells were excluded from analysis based on ToPro3 fluorescence intensity. (D) Representative intracellular calcium oscillations derived from a squared sine function assigned to each sperm in the model simulations. Teal bar at the top of the graph indicates the upper 5% of the concentration range during which the cells were allowed to transition motility states according to a Markov probability transition table. (E) Relative proportion of sperm in each indicated motility state over time (in model-timestep units). In the long run, sperm in the models absorbed into a weak motility state.

function that influences the behavior of the sperm in proportion to the 'frequency' of intracellular calcium transients (Fig 6D). We associated the following equation with each agent in the simulation:

$$\left[Ca^{2+}\right]_i = \frac{1}{2}\left(1 - \cos\Omega_i t\right)$$

Each cell was assigned a randomly generated oscillation frequency $\Omega_i$, drawn from a Poisson distribution. Use of the Poisson distribution was motivated by the skewed indo-1 fluorescence ratio distributions observed in the flow cytometry measurements of intracellular calcium distributions (Fig 6C). It describes the probability of observing a given $\Omega_i$ for an interval of discrete oscillation frequencies with mean $(\lambda)$ and has the property that the sequence of inter-frequency intervals between sperm subgroups will be independent and exponentially distributed with mean $(1/\lambda)$. To model changing population states over time, all sperm began with a progressive motility state and were allowed to change motility states when the intracellular calcium was above $\left[Ca^{2+}\right]_i > 0.97\%$ of the maximum. The changes were stochastic, following a time-homogeneous Markov model, and motility patterns (i.e., progressive, intermediate, hyperactive, slow, and weak) were assigned using a probability transition table (Table 2). Since the weak mobility state was final and could not be left once reached by a given

**Table 2. Table of markov transition probabilities.**

| Current State | To: Progressive | To: Intermediate | To: Hyperactive | To: Slow | To: Weak |
|---|---|---|---|---|---|
| Progressive | 0.95 | .02 | .01 | .01 | .01 |
| Intermediate | .02 | 0.94 | .02 | .01 | .01 |
| Hyperactive | .01 | .02 | 0.94 | .02 | .01 |
| Slow | .00 | .01 | .02 | 0.95 | .02 |
| Weak | .00 | .00 | .00 | .01 | .99 |

sperm, the population experienced a net flow towards this state over a long run period, similar to overall motility degradation observed in live sperm samples (Fig 6E). Overall, this logic models the empirically measured effects of intracellular calcium oscillations on the motility state distributions of sperm [9,28,29], and simulates natural (stochastic) variation in the rates at which individual sperm undergo motility changes during *in vitro* capacitation.

Table: Markov probability transition matrix for changing motility state distributions over time.

## Impact of *phenotype heterogeneity on sperm search*

Next, we simulated search for an egg in microenvironments with increasing spatial complexity (Fig 4A–C). Simulations consisted of 100 sperm, which was chosen as a reasonable (minimal) number that supported consistent random phenotype distributions across simulation runs, determined by increasing the size of sampled distributions until the observed mean approximately matched the ideal mean of the distribution from which the sample was drawn. A two-factor design was implemented to compare the relative effects of sperm population heterogeneity (mean calcium oscillation frequency; $\lambda$) and microenvironmental complexity ($TC_w$) on search time. Intracellular Calcium oscillation frequencies assigned to each sperm were drawn from one of three Poisson distributions characterized by different means/variances ($\lambda$) (Fig 7A). A low $\lambda$ value indicates that most of the sperm had low Calcium oscillation frequencies, and thus, would absorb into a weak motility state slowly, giving them more time to actively search for the egg. Conversely, a high oscillation frequency might absorb into a weakly motile state quickly, making it comparatively less likely to find the egg.

A plot of search time vs. $TC_w$ for each value of $\lambda$ qualitatively indicated that both microenvironmental complexity and phenotypic heterogeneity increased the search time (Fig 7B). Due to the asymmetry of the search time distributions, the assumptions of a two-way ANOVA were not met. To address this issue, we performed a two-way ANOVA on logarithmically transformed search time (Fig 7C), and a statistically significant interaction was detected between $TC_w$ and $\lambda$ ($F_{(4, 891)} = 22.41$; $P < 0.0001$); however, the effect only accounted for 0.29% of total variation. Simple main effects analysis revealed that $\lambda$ accounted for only 0.82% of variation ($F_{(2, 891)} = 127.6$; $P < 0.0001$), while $TC_w$ accounted for most of the variation (95.99%; $F_{(2, 891)} = 14808$; $P < 0.0001$). Post hoc analysis using a Tukey's multiple comparison test indicated statistically significant differences between the $\lambda$ levels in all three microenvironments with mean search time differences as large as ~627 seconds in the most extreme case ($TC_w = 54$; $\lambda = 1$ Hz Vs. $\lambda = 20$ Hz). Together these simulation outcomes predict that both phenotypic heterogeneity and microenvironmental constraints interact to impose selective pressure on fertilizing sperm. An important remaining question is how to quantify both the impact and magnitude of selection on sperm fitness.

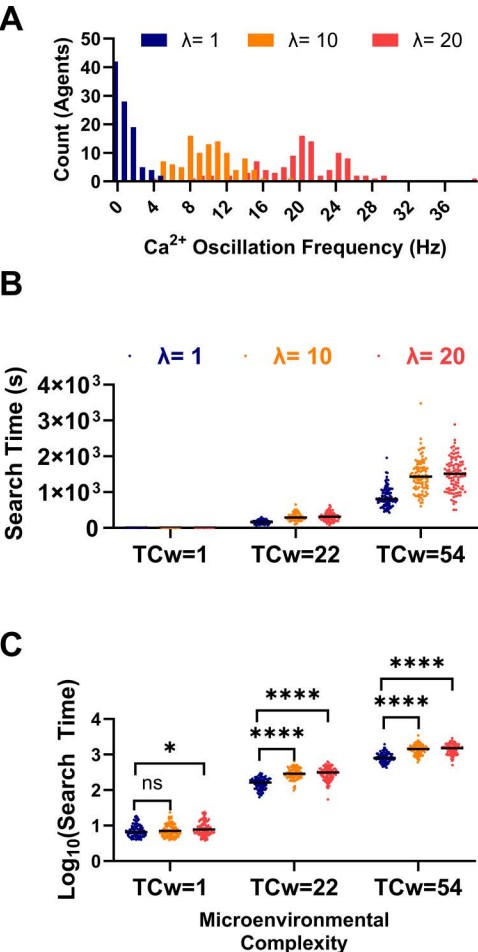

**Fig 7. Impact of Sperm Phenotype Heterogeneity on Diffusive Search.** (A) Histogram of the sperm intracellular calcium oscillation frequencies randomly drawn from a Poisson distribution with the indicated means ($\lambda$). (B) Search time for sperm populations with different phenotype distributions in microenvironments with increasing total weighed complexity (TCw). (C) Logarithmically transformed search times from subplot B used for statistical analysis to satisfy 2-way ANOVA assumptions. Ns = not significant, *p < 0.05, ****p < 0.0001. Lines indicate medians. Simulations consisted of **N** = 100 agents.

## Environmental complexity narrows the posterior distribution of fit sperm

Further exploration of the interaction between sperm-agent population dynamics, spatial constraint, and the effect of selection requires measures of how the distribution of sperm phenotypes changes under selection as well as the magnitude of effect caused by the selective pressure. We considered two measures that have been used in similar biological contexts to model changing phenotype distributions among populations 1.) Bayesian inference - the posterior probability of contact with an egg given that a sperm has a particular phenotype (calcium oscillation frequency $\Omega_i$ in this case), and 2.) Kullback-Leibler divergence (a.k.a. relative information gain) [30] - an information theoretic measure of the *magnitude of effect* of selection on the distribution of calcium oscillation frequencies following diffusive search in microenvironments with differing degrees of complexity. These approaches are especially useful in this context because they are non-parametric and are easily interpreted even when the sample size is small, which is relevant given that only a few sperm may ultimately gain access to the egg during fertilization.

## The posterior distribution provides a quantitative measure of sperm fitness

For these simulations, the agent-based models were updated to facilitate tracking the number, assignment, and duration of contact time with the egg by each of the sperm. A total contact-time threshold of five seconds was defined as a condition to end the simulations. The underlying assumption was that after a threshold value for contacts by one or more sperm, the fertilization process is likely to have occurred if it will occur at all. One hundred simulations involving one hundred sperm each were carried out for each factor and corresponding level (i.e., {$\lambda$: $\lambda$ = 1, 10, 20 Hz}, and {$TC_w$: $TC_w$ = 1, 22, 54}). The relative proportion of the $i^{th}$ Calcium oscillation frequency is denoted $q_i$. To visualize the initial distributions across the $\lambda$ and $TC_w$ levels, cumulative probability distributions were calculated, demonstrating approximately identical initial distributions across each of the one hundred simulations (Fig 8A).

The distribution of each discrete Calcium oscillation frequency among the sperm that contacted the egg is also known as the likelihood function $P(q_i | contact)$. Plotting the likelihood function vs. each distinct Calcium oscillation frequency indicated that increasing environmental complexity narrowed the range of frequencies among sperm that successfully made contact (Fig 8B). It also reduced the absolute number of unique sperm that made contact. Though the likelihood function is the typical empirical measure used in laboratory experiments related to sperm fertility competence, it lacks information about prior assumptions regarding the fitness of sperm traits as well as the base rate distribution of of those traits within the initial sperm population (before fertilization outcomes are known). In other words, the likelihood function is a *sampling* distribution, but what we are most interested in for the purposes of predicting sperm fitness is the posterior distribution, which can be obtained using Bayes theorem:

$$P(contact \mid q_i) = \frac{Prior \ * \ Likelihood}{Total \ Probability} = \frac{P(contact) \ P(q_i \mid contact)}{P(q_i)}$$

where *P(contact)* is the hypothesized prior distribution, in this report $P(contact) = \frac{1}{N}$ ,

where N is the total number of sperm in the simulation. *P(contact)* can be interpreted to mean that all sperm were assumed to have an equal chance of contacting the egg (prior to observing the outcome). Calculating the posterior distribution in this way addresses the question- "what is the probability that a given sperm will be 'successful' given that is has a particular trait value". Cumulative posterior probabilities were calculated for each level of $TC_w$ and $\lambda$ (Fig 8C). Interestingly, the range of successful oscillation frequency values was narrowed by increasing microenvironmental complexity, enabling identification of a subset of sperm that could be considered to have high 'fitness' within each microenvironment.

## Quantifying the magnitude of selection imposed by the microenvironment-

Sperm selection is often used in ART applications, but the magnitude of selective effect is generally not considered, despite the importance of such a measure for comparing the effectiveness of different selection strategies [2,31,32]. Here we describe use of relative information gain as measure of the magnitude of selection. As in the previous section, $q_i$ is the initial distribution of sperm-agent Calcium oscillation frequencies. Let $q_i'$ denote trait probabilities among the proper subset of sperm that successfully made contact with the egg. If $q_i$ and $q_i'$ are the same, then there was no selection for Calcium oscillation frequency during the simulation. However, if $q_i$ and $q_i'$ are not the same distribution, then some subset of Calcium oscillation frequencies did not contact the egg, implying they were selected against by the conditions of

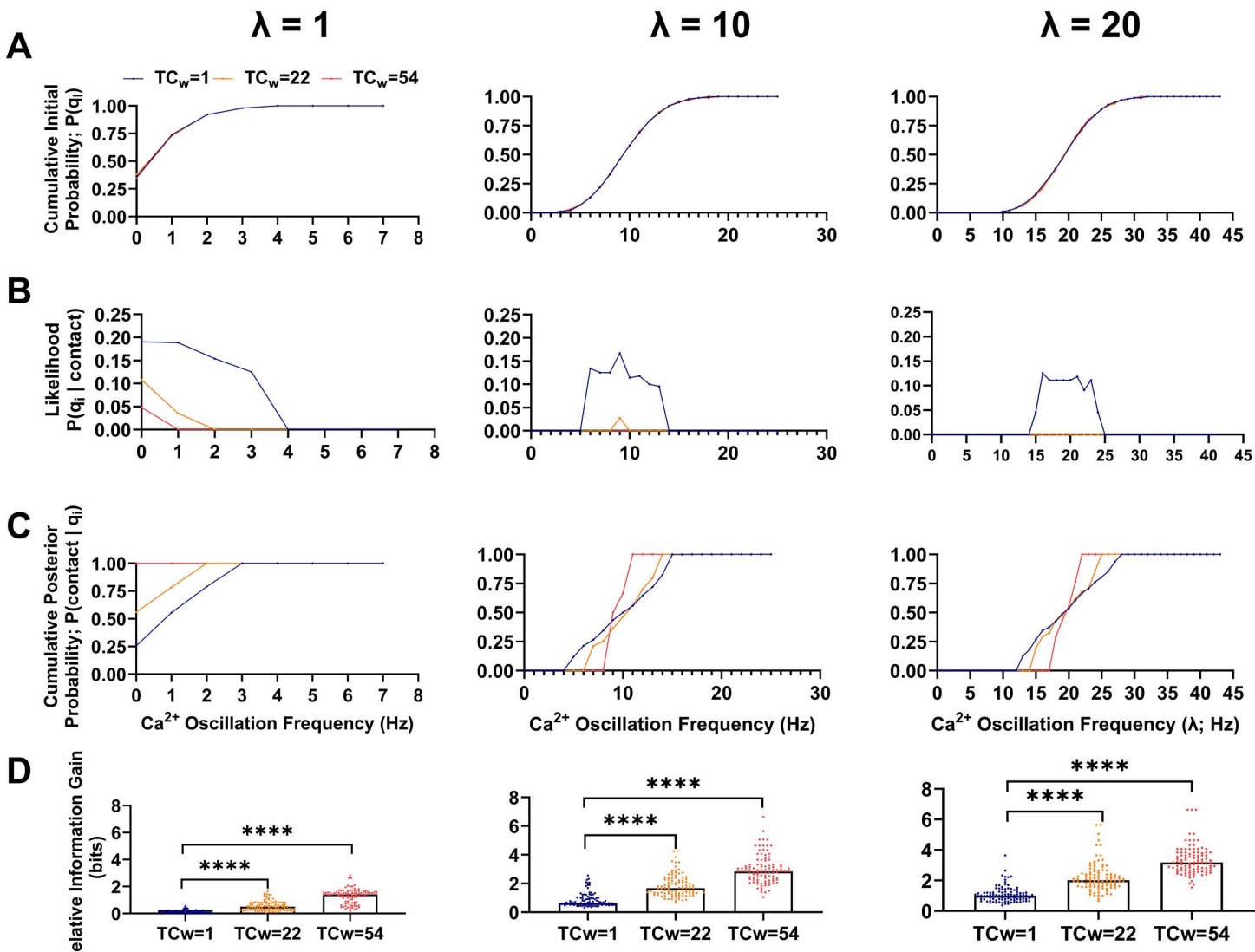

**Fig 8. Measures to Infer Sperm Fitness as well as Quantify the Magnitude of Sperm Selection.** (A) Cumulative distributions total probability P(qi) for each oscillation frequency in the initial sperm population for each simulation condition. (B) The Bayesian likelihood (frequency of sperm for each oscillation frequency that contacted the egg). (C) Cumulative posterior probability of egg contact for each oscillation frequency. Note, a prior distribution of 1/N, where N is the total number of sperm in the simulation, was used in the calculation. This can be interpreted to mean that each sperm had an assumed equal chance of contacting the egg. (D) Relative information gain (a.k.a. Kullback-Leibler divergence) calculated for each simulation condition. ****p < 0.0001. TCw = total weighted complexity. **N** = 100 sperm in each simulation. Points in A-C represent the median of 100 simulations. Points in D represent relative information gain for each of 100 simulations.

the simulation. The normalized distance between the two distributions can be quantified with the following expression:

$$D(q' \;||\; q) = \sum_{i=1}^{n} q_i' \log_2\left(\frac{q_i'}{q_i}\right)$$

This expression is known as the relative information gain (or Kullback-Leibler divergence), and its units are in binary digits (bits). A relative information gain of 0 indicates the distributions are the same, and a positive number indicates the magnitude of the difference between the two distributions. The relative information gain cannot be less than 0 and this measure is

not symmetric, meaning that it is not equivalent to $D(q \| q')$. As anticipated, increasing $TC_w$ or $\lambda$ increased the relative information gain (Fig 8D). Two-way analysis of variance (ANOVA) revealed a statistically significant interaction effect among $TC_w$ and $\lambda$ (F (4, 891) = 19.47; P < 0.0001), which only accounted for ~2% of the total variation. Simple main effects analysis indicated that $TC_w$ accounted for about 37% of the variation (F (2, 891) = 542.9; P < 0.0001) and $\lambda$ accounted for about 29% (F (2, 891) = 425.9; P < 0.0001). The results of a Tukey's post hoc test comparing each $\lambda$ level to $\lambda = 1$ is indicated in (Fig 8D).

Taken together, the results from these simulations reinforce the conclusion that simple spatial hindrance by the latent structure of the microenvironment combined with variation in individual sperm phenotypes exerts quantifiable selective pressure on sperm during fertilization. The proposed measures enable quantitative description of sperm fitness and the magnitude of selection in relatively simple terms.

## Discussion

### Random walks

The asymptotic properties of diffusive and correlated random walks have been well studied and described previously [33]. The goal of this report is not necessarily to derive new findings on random walk behavior, but rather, to frame the physiological context of the sperm search for an egg using tools from this field with the hope of illuminating improved fertility analysis and/or prediction. General random walk models, similar to those presented in this report, have several notable properties- 1) First passage probabilities, akin to the probability that a sperm will find an egg, are heavily influenced by dimensionality [34], 2) correlation among successive steps (i.e., persistence of the walkers), akin to progressive sperm motility patterns, produces scaling properties that are distinct from uncorrelated random walks [34], 3) the number of distinct sites visited by a given number of random walkers, akin to how sperm explore space, has been shown to exhibit distinct time regimes that depend on the system size (i.e., the area of the environment and the number of random walkers) [35] and finally, 4) random walks on undirected graphs, akin to sperm movement in the epithelial folds of the uterus and oviducts, exhibit polynomial traversal times- bounded above by $T \leq 2e(n-1)$, where $e$ is the number of edges and $n$ is the number of nodes [23]. Together, these properties may be extended to inform the behavior of diverse sperm populations searching in spatially complex microenvironments.

*A Systems Perspective on Fertilization* - The molecular mechanisms that underpin the regulation of mammalian sperm post-ejaculatory maturation (a.k.a., capacitation) have been thoroughly studied, and consist of signaling pathways [26,36], metabolic processes [24,25,37], and complementary binding of cell surface molecules [38,39]. Though there are excellent physical models of individual sperm motility function and regulation [16,18,40–43], few models account for the stochasticity of hundreds of millions of sperm searching for an egg that ultimately determine fertility outcomes; this gap in knowledge persists despite consistent observations of significant phenotypic heterogeneity within sperm populations (e.g., the localized expression of ion channels on the plasma membrane) [4,6,9,44].

Previous work modeling sperm search times in both 2D and 3D environments detailed several potential scaling laws for the diffusive search process relating search time to sperm number [45]. Similar to our observations in this report, a non-linear relationship between sperm number and search time was described, and the scaling relationships depended on the dimensions of the search space. However, those simulations used sperm with constant velocity, rectilinear motion, and did not explicitly account for phenotypic heterogeneity or motility pattern changes over time. The agent-based models (ABMs) developed in this report

are informed by empirical data and provide a structured framework to explore the complex collective dynamics of phenotypically heterogeneous sperm populations under various environmental conditions. The models facilitate a deeper understanding of the interactions between microenvironmental complexity and sperm phenotypic heterogeneity, emphasizing the stochastic nature of the variables that shape sperm fitness.

### Diffusive search under microenvironmental constraint

The spatial scale of motility is important when analyzing the consequences of motility pattern distributions on sperm selection. *In vivo*, peristaltic fluid flow moves sperm suspensions over relatively large distances within the female reproductive tract independent of their motility status [21]. Though this phenomenon will distribute the cells on a macroscopic scale, at the microscopic scale, individual sperm must still 'search' local space using flagellar movement in a manner that increases probability of contact with the egg. This important property implies that critical cellular density thresholds, cell intrinsic motility characteristics, and microenvironmental factors such as physical/chemical barriers play critical roles in influencing which sperm from a given cell population will have an opportunity to fertilize. The degree to which this is due to chance alone is an important consideration, and a theoretical framework for sperm selection should account for the probabilistic dependencies of sperm fitness. The simulations in this report predict that increased microenvironmental complexity requires greater sperm density to maintain effective diffusive search and timely egg contact, highlighting potential tradeoffs between the collective diffusive search capability of a sperm population and the number of required sperm. The models also predict that critical thresholds exist, above which sperm number plays a diminishing role in diffusive search capability. Importantly, this threshold is not a fixed value, but rather, depends on the complexity of the microenvironment and the phenotypic heterogeneity of the sperm population.

### Impact of sperm phenotypic heterogeneity

Motility is the most fundamental physiological function of mammalian sperm and is a common distinguishing feature used in clinical sperm selection [31,32]. Our results demonstrate how intrinsic phenotypes of sperm, such as intracellular ion transients coupled with the regulation of motility pattern, may critically influence selection outcomes. Sperm phenotypic variation is complex and caused by many different factors. Variation may be an important driver of optimal sperm number among (or within) species, based on the observation that sperm number is positively correlated with potentially deleterious effects of genomic recombination during meiosis [46]. Variation may also be undergirded by an evolutionarily stable strategy that optimizes the number of capacitated sperm during a post-copulatory fertilization window; a process facilitated by periodic synchronous capacitation among sperm subpopulations [47]. Regardless of the underlying causes of variation, the simulations in this report suggest that the reproductive microenvironment is a critical factor in sperm selection because it ultimately determines which sperm phenotypes will have access the egg. This suggests that sperm selection protocols for ART should consider both the statistical distribution of biological variability among the sperm and the physical/chemical structure of the microenvironment in which fertilization will occur.

### Quantifying sperm fitness and selection pressures

Sperm pre-selection is almost ubiquitous in routine clinical diagnostics and ART procedures (e.g., gradient centrifugation, swim-up assays, hyaluronan binding assays, etc.).

Though semen parameters such as motility and sperm count are known to influence ART outcomes [48], current methods of selection largely depend on simple *correlation* and qualitative assumptions about the effects of selection [49]. Additionally, use of ICSI has increased substantially in recent decades, a procedure which relies on direct selection of a single sperm for injection into the egg [50]. Currently, there is no quantitative framework that facilitates high-precision sperm selection from *within-male* samples and sperm 'fitness' remains nebulously defined.

Though fitness could be quantitatively framed in many ways - Bayesian inference is a particularly useful approach because it is insensitive to the base rate representation of sperm phenotypes and has no minimum sample size. The approach taken here is drawn from the theory of natural selection, in which fitness is defined as a probability measure of success [51], which has classically been defined by survival or reproduction of organisms within a given population. For the purposes of modeling fertilization, success can be defined flexibly depending on the scenario without altering the underlying mathematical representation (e.g., egg contact, fertilization, passage through a selective barrier, etc). Most approaches to assessing sperm fitness are based on regression of sperm traits (or interventions) with fertility or developmental outcomes [52,53]. However, it is important to consider that the probability that a sperm has a particular trait given that it fertilized an egg, is not necessarily equivalent to the probability that a sperm *will fertilize* an egg given that it has a particular trait, though it is the latter condition (inference) that is of prime interest for sperm selection in ART.

Bayes theorem incorporates useful information beyond simple sampling frequencies. For example, it accounts for the relative proportion of sperm with 'successful' sperm traits in the initial population and prior information about the traits' contributions to fertilizing potential. Bayesian inference has been used recently in conjunction with dimensionality reduction to make fertility predictions from motility stereotypes in boar semen [54]. Quantitatively defining sperm fitness is becoming more important as machine learning and computer vision technologies advance, allowing for high-dimensional data collection from semen samples, and necessitating methods that analyze distributions directly rather than relying on summary statistics [55]. One advantage of the approach taken in this report is the computational simplicity, which may be useful for developing classification or selection strategies that rely on interactive microscopy video manipulation in real-time.

Another major limitation to improving current male fertility diagnostics and high-precision selection is that fertilization is an open-ended process, making it very difficult to predict which sperm will have a selective advantage from semen analysis alone. As mentioned previously, the microenvironment plays a substantial role in constraining which sperm will have access to the egg. For this reason, it is critical to have some measure that can compare between the selective effects introduced by different reproductive microenvironments *in vivo* or *in vitro*. To address this limitation, we propose another measure - relative information gain - for quantification of the magnitude of selection imposed by the reproductive microenvironment [51]. This measure, also known as Kullback-Leibler divergence, provides a useful way to compare selection strategies quantitatively [30]. It is a numerical representation of the 'distance' between the trait distribution of an initial (pre-fertilization) population of sperm and the 'successful' (post-fertilization) population of sperm. Notably, it is independent of the actual features of the microenvironment and is only dependent on the effect of microenvironmental constraint on sperm fitness. Additionally, it lays the groundwork for a new biological context for reproduction by reframing the process of fertilization in information theoretic terms as a form of learning process [56].

## Model limitations

There are several notable limitations of the models developed in this report. Most of the limitations stem from the simplifying assumptions made about the physiological phenomena that underly mammalian reproduction. First, the sperm movement functions are relatively simple, but real sperm exhibit more complicated patterns such as helical progression [57]. Second, the regulatory systems in the model that control the timing and trajectory of capacitation are limited only to calcium transients with an assumed correlation between intracellular calcium concentrations and motility pattern transitions. This simplification ignores a much more complicated reality involving time-inhomogeneous plasma membrane potassium hyperpolarization, metabolic energy balance, protein tyrosine phosphorylation, and other key biochemical reactions. Though the results should be interpreted with caution, the models were designed to capture key elements of cell population scale dynamics and were constrained by empirical data to enhance their physiological relevance. These models may be extended to incorporate updated movement parameters and regulatory subsystems - for example through use of coarse-grained approaches such as Boolean networks or more involved systems of differential equations [9,58]. Finally, to approach this problem in a general way, we modeled only *non-adaptive* sperm, meaning sperm that do not change their motility pattern in response to environmental inputs. However, there are many ways by which mammalian sperm modulate behavior in response to their environment including chemotaxis, rheotaxis, and thermotaxis. Incorporating these behaviors will likely affect the statistical predictions about sperm fitness and should be pursued in future studies.

## Materials and methods

### Ethics statement

All animal related work adhered to the guidelines outlined in the National Research Council Guide for the Care and Use of Laboratory Animals and was approved by the Institutional Animal Care and Use Committee of East Carolina University (approval A3469-01).

### Model implementation

Agent-based models were developed and implemented using the Netlogo modeling environment (V6.2.2) [59]. Netlogo BehaviorSpace was used for repeated simulations with parameter scaling. The models and other supporting information are available at (https://github.com/cas-mitolab/Fertilization_ABM). Simulations were run on a standard laptop computer with 16 GB of RAM and an Intel Core i7 1.7GHz processor. Markov state transition simulations and calculations related to Bayesian inference and relative information gain (Kullback-Leibler divergence) were performed using the Python (V3.9) programming language and the NumPy library (https://github.com/cas-mitolab/Fertilization_ABM) [60].

### Animals

Adult male outbred CD-1 retired breeder mice were obtained from Charles River Laboratories (Raleigh, NC, USA). Mice had free access to water and food, were maintained on a 12-hour light/dark cycle and were humanely euthanized by CO2 asphyxiation followed by thoracotomy.

### Isolation of mouse epididymal sperm

Testes with epididymides were isolated in phosphate buffered saline (PBS) at 37°C. Cauda epididymides were transferred to isolation media where gently dissected. Following a brief

swim-out period (~15 minutes at 37°C), sperm were isolated from epididymal tissue by centrifugation at 100 x g for 2 minutes. Cell counts were determined using a hemocytometer after dilution in water. Cells were then incubated at 37 °C for 30 minutes with 10 μM Indo-1-AM cell permeable free-calcium dye in HEPES buffered, bicarbonate-free, media containing glucose and lactate (2.88 mM and 21 mM respectively). Cells were then washed by centrifugation at 800 x g for 5 minutes.

## Calcium clamp and microtiter plate assay

1 mM EGTA (ethylene glycol-bis(β-aminoethyl ether)-N,N,N′,N′-tetraacetic acid) was used to clamp the 'free' Calcium ion concentration in assay preparations. Calcium concentrations were measured using ion selective electrodes (Kwik Tip electrodes; World Precision Instruments, Sarasota Fl, USA). Two concentration ranges were determined, requiring two separate electrode filling solutions (low range - 150-300 μM; high range - 1.2-2.0 mM) with different concentrations of $CaCl_2$ to obtain an appropriate working range. Once determined, the buffer conditions that clamped the Calcium concentrations were included in the assay in conjunction with sodium bicarbonate pseudo-titrations. pH of the media during the assays did not change and was confirmed using glass tipped pH microelectrodes (World Precision Instruments, Sarasota Fl, USA). Indo-1 stained cells were added into the microtiter plate at uniform cell density across the plate and fluorescence (340/400:475 nm) was obtained for ratiometric analysis at 37 °C with sampling every 5 minutes for two hours in a microtiter plate reader (Molecular Devices ID3, San Jose CA, USA). The calcium ionophore ionomycin (10 μM) was used as a positive control.

## Spectral flow cytometry

– The qualitative distribution of intracellular calcium in live sperm populations was performed using a 5-laser Aurora spectral analyzer with SpectroFlo acquisition software (V2.2; Cytek, Fremont, CA, USA) [61]. Flow cytometry measurements were performed in capacitating media with corresponding pseudo-titrations of calcium chloride and sodium bicarbonate. Scatter gating was used to identify intact single cells. Live-cell impermeable ToPro3 dye (Thermo Fisher; Waltham MA, USA) was used to monitor cell viability during the assays and mild detergent titrations (digitonin, 24-240 μM) were used to prepare single stain reference for dead cells. Indo-1-AM was excited using a 405nm laser and emission collected at 400 and 475 nm. Conditions were optimized prior to flow cytometry by spectral scanning using a Horiba Duetta fluorometer (Kyoto, Kyoto, Japan). Ionomycin was included as a positive control condition. Scatter plots of fluorescence intensity were manually gated and exported using FlowLogic (V8.7, Inivai Technologies; Victoria AUS). Kernel density estimates of fluorescence ratios for various pseudotitration conditions were plotted using Python (V3.9) with Matplotlib, NumPy, and Pandas libraries.

## Data analysis and statistics

Data were analyzed and visualized using Graphpad Prism (V9.1.2), or NumPy, Pandas, and Matplotlib [60,62,63]. Statistical analyses were performed using Graphpad Prism (V9.1.2, San Diego, CA, USA). Two tailed Student's t-test was used for comparison of group means. Normal quantile-quantile plots were used to assess whether normal based inference procedures should be replaced with nonparametric methods. The presence of outliers, both their magnitude and number, was also used to check the assumptions of inference procedures. For multifactorial designs one- or two-way ANOVA was performed for one or two factor designs, with Dunnett or Sidak *post hoc* tests for multiple comparison respectively. All data are presented as

raw values with the median represented by a bar. An α value of 0.05 was used as the threshold of statistical significance.

## Conclusion

In this report we developed agent-based models (ABMs) and explored aspects of collective behavior of non-adaptive sperm (i.e., sperm that change motility pattern over time in a manner that depends on intrinsic control, rather than exogenous responses to signals). Our results highlight the intertwined influences of microenvironmental complexity and sperm phenotypic heterogeneity in shaping sperm fitness- defined here as the probability of egg contact given that a sperm has a particular trait value. Results from this study provide key insights and useful definitions for further exploration of a theory of sperm selection in the context of assisted reproductive technologies. The insights provided by the models hold promise for optimizing real-time sperm diagnostics and selection strategies with broad applications in both clinical and agricultural settings.

## Author contributions

**Conceptualization:** Benjamin M. Brisard, Kylie D. Cashwell, Stephanie M. Stewart, Logan M. Harrison, Aidan C. Charles, Chelsea V. Dennis, Ivie R. Henslee, Ethan L. Carrow, Heather A. Belcher, Debajit Bhowmick, Paul W. Vos, Maciej Majka, Martin Bier, David M. Hart, Cameron Alan Schmidt.

**Data curation:** Benjamin M. Brisard, Kylie D. Cashwell, Stephanie M. Stewart, Logan M. Harrison, Aidan C. Charles, Chelsea V. Dennis, Ivie R. Henslee, Ethan L. Carrow, Heather A. Belcher, Debajit Bhowmick, Paul W. Vos, Maciej Majka, Martin Bier, David M. Hart, Cameron Alan Schmidt.

**Formal analysis:** Benjamin M. Brisard, Kylie D. Cashwell, Stephanie M. Stewart, Logan M. Harrison, Aidan C. Charles, Chelsea V. Dennis, Ivie R. Henslee, Ethan L. Carrow, Heather A. Belcher, Debajit Bhowmick, Paul W. Vos, Maciej Majka, Martin Bier, David M. Hart, Cameron Alan Schmidt.

**Funding acquisition:** Cameron Alan Schmidt.

**Methodology:** Cameron Alan Schmidt.

**Project administration:** Cameron Alan Schmidt.

**Validation:** Cameron Alan Schmidt.

**Visualization:** Cameron Alan Schmidt.

**Writing – original draft:** Benjamin M. Brisard, Kylie D. Cashwell, Maciej Majka, Cameron Alan Schmidt.

**Writing – review & editing:** Benjamin M. Brisard, Kylie D. Cashwell, Maciej Majka, Cameron Alan Schmidt.

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
