## [Decision Letter · Decision Letter 0]

7 Oct 2024

Dear Dr Schmidt,

Thank you very much for submitting your manuscript "Modeling Diffusive Search by Non-Adaptive Sperm: Empirical and Computational Insights" for consideration at PLOS Computational Biology.

As with all papers reviewed by the journal, your manuscript was reviewed by members of the editorial board and by several independent reviewers. In light of the reviews (below this email), we would like to invite the resubmission of a significantly-revised version that takes into account the reviewers' comments.

We cannot make any decision about publication until we have seen the revised manuscript and your response to the reviewers' comments. Your revised manuscript is also likely to be sent to reviewers for further evaluation.

Sincerely,

Jing Chen

Academic Editor

PLOS Computational Biology

Pedro Mendes

Section Editor

PLOS Computational Biology

Reviewer's Responses to Questions

**Comments to the Authors:**

Reviewer #1: uploaded as an attachment

Reviewer #2: In this manuscript, the authors suggest to use simple modelling approach to study the selective pressure and variability in the sperm fitness during the fertilisation process. I actually do like the general idea and fully support the usage of simple enough models to make the point. Indeed with the help of tractable models it is possible to bring across very powerful mathematical predictions that can be the key to understanding even such complex phenomena as discussed in this manuscript. So while in general strongly supporting the simple idea, I have multiple technical comments/problems to the implementation which prevent me from being fully positive about this work. Below I provide the comments in the chronological order. I note the most critical points with a star * sign. Also there is a tendency to over-interpret the results from the models

- “Diffusive search is an intrinsic property of agents performing a random walk” - this is a strange statement, not all random walking agents are searching for something

- Fig 1 A is confusing, not clear what the blue, red and white parts are, also what is the zoom in square means. Maybe showing a photo of a real chip would help better than this schematics

- Angle \theta in Fig. 1B is just showing the 90 degrees and not linked to the trajectory

- "increasing the mean \mu .." - mean of what?

-"Minor adjustments to the mean or standard deviation of the distribution can have significant effects on the agents' diffusive search behavior" - this is very vague and unsubstantiated statement. What is minor, what is significant? Also search behaviour is not quantified in any way.

-"Agents with movement parameters that allow them to search space more quickly than others will exhibit a more rapidly increasing RMS” - this is also very vague statement. A walker moving along the straight line with constant speed will have higher asymptotic MSD - but will it search space more quickly? It is very imprecise formulation.

-How random \delta is selected? can it be negative (if it comes from Gaussian distribution)?

-“..search patterns to broader exploration of the surrounding environment ultimately resulting in altered diffusive search outcomes” - this is either unsubstantiated statement or trivial, should be removed in both cases.

(*)-“Those with smaller ranges had fewer directions to go at each timestep and exhibited behavior that resembled crude progressive motility patterns (Figure 1E)” - how do we know? no reference, no figure

-" Taken together, the results of these simplified models highlight the diffusive search functionality that emerges” - unsubstantiated - nothing definite has been said about search till this point.

-"match the simulated Gaussian distributions from data” - not clear what it is, why to simulate Gaussian from data?

-Fig 2A: why experiment and simulations are shown at different magnification? It is impossible to see the tracks in simulations good enough to say if they are similar or not

(*)- "The resulting sperm motility quantitatively and qualitatively like mouse sperm..” how do we see that? based on which analysis? We see in Fig. 2 BCD that multiple quantifiers are off between data and simulations. Some have low values and we can’t tell how good/bad they are

- "These results demonstrate that the average diffusive search capability of a sperm population reflects the underlying distribution of sperm phenotypes. In other words, the ability to efficiently search space is an emergent property of the sperm population.” This is somehow very generic and trivial statement. The search efficiency was quantified with an arbitrary quantity (which is not really discussed who general it is) and then of course - different microscopic random walk leads to a different diffusion constant and to different exploration - of course? what else?

- The plots for the MSD, within the realm of the random walk model used can be fully analytically quantified, in particular by the MSD of the Ornstein-Uhlenbeck process that can be derived from the run-and-tumble-like random walk model.

-"The simulations presented thus far predict that sperm populations will function on average to search all the space that they occupy, and the kinetics of the search process depend on the distribution of sperm motility patterns within the cell population” - There is nothing to predict! The result is known (trivial) analytically as soon as the model is formulated? Rephrase!

-Figure 5. I have difficulties with quantitative interpretation of these results, as the space/time scale of the experiment is unclear, and very difficult to tease out from the methods section. Also form that moment on some quantities start to be shown stochastically with units/dimensionless.

(*)-Results in Figure 5 in essence have clear, analytical explanation. For the open chamber the search time is the first passage time of the random walk to the target. And the plots show how that changes for a finite number of trajectories. The shortest possible time is the movement of the random walker on a straight path (very rare event). For an infinite number of realisations such a trajectory will always appear and thus provide a narrow cloud of points for 100 trials - this is an asymptotic value towards which the times converge. This behavior is to be expected as soon as the number of trajectories (particles) times the statistical probability of such trajectory to occur (can be calculated, like the authors did for labyrinths) will become equal to 1 - so one trajectory is enough. For smaller N of course we have to wait till whatever trajectory first reaches the target (thus a scatter and higher target hitting time).

Similarly - plots in B show the interplay of the time it takes to reach the target versus the number of distinct sites visited by the random walker. Reaching the target depends on the pdf of the particles to reaching out beyond a given distance with high enough total probability. In that there is a square root relationship of the distance and the time it takes to reach the target (if we argue in terms of time scaling). Another quantity is the number of distinct sites visited by the collection of random walkers. The scaling of this quantity for large N (number of walking particles) has been studied in 90s (see for example Larralde et al PRA 1992). The number of distinct sites visited with time (depending on the regime) scales at least linearly. So therefore it is also expected (from analytical arguments) that the number of sites visited with the increasing number of walkers will increase reaching to the 100% before the run is terminated when one of the particles hits the target.

- "Taken together, these results predict that diffusive search by non-adaptive sperm will exhibit a non-linear relationship with sperm density and that the role of chance in finding shortest path to the egg is modified by the spatial complexity of the microenvironment”. So, as before, I don’t think there is any need in the numerical simulations to predict the behaviours observed/shown in the plots. The qualitative outcome can be understood based on the known results from basic random walk properties. The only novelty (which is also probably difficult to straightforwardly derive analytically) are the results for labyrinth geometries. Those however, have no qualitative explanations, for example, with respect to the characteristic time scales, etc. So these are not predictions, but a simple numerical verification of the analytically predictable (interpretable) results.

- "with a Markov transition matrix “ - this is not defined, also in the Methods it is hard to understand how exactly it looks like

- "Simulations consisted of 100 sperm, which was chosen as a reasonable (minimal) number” - based on what?

-" with mean search time differences as large as ~627 seconds in the most extreme case” as mentioned above, the jumping between unit and unitless quantities - and no idea of the actual searching area - it is hard to give a physical meaning to the numbers provided.

(*) - "Taken together, the results from these simulations reinforce the conclusion that simple spatial hindrance by the latent structure of the microenvironment combined with variation in individual sperm phenotypes exerts quantifiable selective pressure on sperm. The fitness can be represented as an inferential probability of ‘success’, and the magnitude of selection can be represented using relative information gain.” I think this is the part which mixes the relatively straightforward predictions of the model (which are essentially what they are as soon as parameters are fixed) and the pressure/selection are the criteria imposed by the authors, and the more complex evolutionary concepts of the underlying biological problem. Instead, what authors should do - just explain their mathematical results, and then map them in the language of pressure and selection, but not mixing them (as also was done in the whole last section).

"Previous work modeling sperm search times in both 2D and 3D environments detailed several potential scaling laws for the diffusive search process relating search time to sperm number[38]” - the work cited, which is indeed based on a large body of analytical work, actually argue that the relevant movement of cells is not diffusive as considered in this work but along the straight paths which only reflect from the boundaries of the confinement. This needs to be commented.

" The simulations in this report predict that increased microenvironmental complexity requires greater sperm density to maintain effective diffusive search and timely egg contact” - as mentioned in the above several points - these are not the points that need any numerical simulations to make predictions. At best those are numerical illustrations, and maybe the way to get a feeling of the quantitative effect.

- My final point is that in the extensive discussion section there are no qualitative explanations of the model results, which, in fact, can be argues in terms of the random walk theory. In terms of numbers (times/scales) there is no discussion of relevance to those to the underlying biological problem. Furthermore, the connection of the model results to the problem of selection and pressure need a dedicated explanation, why the quantifiers/measure are the one to use? What are the alternatives, what are the advantages/disadvantages?

So taken together, as you see, there is an extensive list of criticism on the technical implementation and its interpretation which need to be resolved before reconsidering this manuscript for publication.

**Have the authors made all data and (if applicable) computational code underlying the findings in their manuscript fully available?**

Reviewer #1: Yes

Reviewer #2: **No: ** No access to code, some descriptions of the model setup are still cryptic.

PLOS authors have the option to publish the peer review history of their article (what does this mean? ). If published, this will include your full peer review and any attached files.

**Do you want your identity to be public for this peer review?** For information about this choice, including consent withdrawal, please see our Privacy Policy .

Reviewer #1: No

Reviewer #2: No
---

## [Decision Letter · Decision Letter 1]

9 Jan 2025

PCOMPBIOL-D-24-01402R1

Modeling Diffusive Search by Non-Adaptive Sperm: Empirical and Computational Insights

PLOS Computational Biology

Dear Dr. Schmidt,

Thank you for submitting your manuscript to PLOS Computational Biology. After careful consideration, we feel that it has merit but does not fully meet PLOS Computational Biology's publication criteria as it currently stands. Therefore, we invite you to submit a revised version of the manuscript that addresses the points raised during the review process.

Please submit your revised manuscript within 30 days Mar 11 2025 11:59PM. If you will need more time than this to complete your revisions, please reply to this message or contact the journal office at ploscompbiol@plos.org. Please include the following items when submitting your revised manuscript:

We look forward to receiving your revised manuscript.

Kind regards,

Jing Chen

Academic Editor

PLOS Computational Biology

Pedro Mendes

Section Editor

PLOS Computational Biology

**Additional Editor Comments:**

Please address the reviewers' new comments regarding the clarity issue of the revised manuscript. An accurate description of the contribution of this work and justification of the methods will make the work more significant and impactful.

**Journal Requirements:**

1) We have noticed that you have uploaded Supporting Information files, but you have not included a complete list of legends. Please add a full list of legends for your Supporting Information files after the references list.

2) Please amend your detailed Financial Disclosure statement. This is published with the article. It must therefore be completed in full sentences and contain the exact wording you wish to be published.

**Reviewers' comments:**

Reviewer's Responses to Questions

**Comments to the Authors:**

Reviewer #1: The paper presents computational results that support empirical methods for assessing sperm fitness within assisted reproductive technologies. The authors conceptualize the physiological function of sperm as a diffusive search process and develop computational tools to explore the underlying causal dynamics of sperm fitness. They introduce both a probabilistic measure of sperm fitness and an information-theoretic measure of sperm selection, each situated within a relevant theoretical framework. The figures, methods, and explanations collectively support the idea that the developed agent-based models present compelling arguments, offering valuable insights toward a theory of sperm selection. Overall, the references and detailed results are consistent and well-explained. The responses to reviewer comments have strengthened the paper's main mathematical results and established stronger connections with relevant empirical work.

The comments raised by reviewers were adequately addressed, and the main issues have been resolved. Below are additional comments and questions regarding the responses to the original reviewer feedback:

Comments on the GitHub Repository Updates

From Document:

"Response: We have updated the movement function description in the paper to make the model implementation more clear as well as to address potential issues with compounding random noise in the initial models. The movement functions are described in the materials and methods. We have modified Figure 1 and no longer include the simulations with simple Gaussian walks in lieu of simulations with the updated movement functions. We have updated the code in the GitHub repository accordingly."

Comment:

Although the responses indicate that the code source at https://github.com/CAS-ReproLab/Fertilization_ABM has been updated, the GitHub repository itself has not received any updates in the past six months. It is possible that updates have been made to a nonpublic version of the code but have not yet been pushed to the public repository. Please clarify if there is a new link or specify which codes have been updated.

Comments on Future Work and Current Paper Content

From Document:

"Response: Not sure I fully understand this critique, so apologies in advance if the response is off base. In sperm analysis, summary statistics are often used to describe sperm phenotypes (e.g., motility measures, etc.) without paying much attention to the underlying distributions, which in our experience are often not symmetric. We feel that this perspective on sperm physiology misses the point that the underlying distributions matter greatly, and ignoring this likely contributes to diagnostic shortcomings and misconceptions about the physiological tasks that sperm perform. The approach to quantifying fitness and the magnitude of selection may be an important step toward improving sperm diagnostics and understanding the constraints that influence which sperm will ultimately fertilize an egg. We realize that the idea is mostly descriptive and preliminary, but we are working on a 'more detailed analysis of the underlying statistical considerations as well as the information-theoretic implications of selection during fertilization.'”

Comment:

The more detailed analysis that is being worked on, I am assuming that is referring to a future work. For this comment it would be useful to address the current information theoretic sections of the paper to make clear the distinction between what is present in the paper and the future proposed work. This distinction would both enhance the reader’s understanding of the scope and implications of the current research and ensure the reviewers are adequately understanding the points that you are getting across.

Reviewer #2: While the authors did a good job in removing ambiguous passages and overstatements the presentation has now suffered in two ways. First when reading the results, it is not clear what the setup and setting is, what are the mazes ABC where the egg is, what the sperm is doing while moving in the model. I really do not understand why this information is now hidden in S1 figure (which still doesn’t contain mazes). Second negative effect is that by removing the overstatements there is now no any kind of qualitative explanation of the results (at least in the way I was trying to make sense of those results in my previous comment). So now we are only presented with some results (dependencies) but no attempt to rationalize them.

Taken together, while the over interpretations were removed, the clarity of results has suffered considerably. I do think this needs to be fixed before it can be published.

**Have the authors made all data and (if applicable) computational code underlying the findings in their manuscript fully available?**

Reviewer #1: Yes

Reviewer #2: Yes

PLOS authors have the option to publish the peer review history of their article (what does this mean? ). If published, this will include your full peer review and any attached files.

**Do you want your identity to be public for this peer review?** For information about this choice, including consent withdrawal, please see our Privacy Policy .

Reviewer #1: No

Reviewer #2: No

**Figure resubmission:**
---

## [Decision Letter · Decision Letter 2]

10 Feb 2025

Dear Dr Schmidt,

We are pleased to inform you that your manuscript 'Modeling Diffusive Search by Non-Adaptive Sperm: Empirical and Computational Insights' has been provisionally accepted for publication in PLOS Computational Biology.

Best regards,

Jing Chen

Academic Editor

PLOS Computational Biology

Pedro Mendes

Section Editor

PLOS Computational Biology

Please fix one small type on line 412: of of.

Reviewer's Responses to Questions

**Comments to the Authors:**

Reviewer #1: Overall, the references and updated sections are consistent and well-explained. The responses to comments from reviewers have further strengthened the main mathematical results of the paper along with building a stronger connection between relevant empirical works. Overall, the comments were fully addressed and the main issues brought up were resolved.

Reviewer #2: I appreciate that the authors put the details of the model back to the main text. I reread the whole manuscript again and it now reads much more consistently. I’m happy to recommend the paper for publication.

I only noticed one small type on line 412: of of

**Have the authors made all data and (if applicable) computational code underlying the findings in their manuscript fully available?**

Reviewer #1: Yes

Reviewer #2: Yes

PLOS authors have the option to publish the peer review history of their article (what does this mean? ). If published, this will include your full peer review and any attached files.

**Do you want your identity to be public for this peer review?** For information about this choice, including consent withdrawal, please see our Privacy Policy .

Reviewer #1: No

Reviewer #2: No

---

## [Editor Report · Acceptance letter]

PCOMPBIOL-D-24-01402R2

Modeling Diffusive Search by Non-Adaptive Sperm: Empirical and Computational Insights

Dear Dr Schmidt,

I am pleased to inform you that your manuscript has been formally accepted for publication in PLOS Computational Biology. Your manuscript is now with our production department and you will be notified of the publication date in due course.

With kind regards,

Zsofia Freund
